# End-to-End Differentiable Proving

**Tim Rocktäschel**
University of Oxford
tim.rocktaschel@cs.ox.ac.uk

**Sebastian Riedel**
University College London & Bloomsbury AI
s.riedel@cs.ucl.ac.uk

## Abstract

We introduce neural networks for end-to-end differentiable proving of queries to knowledge bases by operating on dense vector representations of symbols. These neural networks are constructed recursively by taking inspiration from the backward chaining algorithm as used in Prolog. Specifically, we replace symbolic unification with a differentiable computation on vector representations of symbols using a radial basis function kernel, thereby combining symbolic reasoning with learning subsymbolic vector representations. By using gradient descent, the resulting neural network can be trained to infer facts from a given incomplete knowledge base. It learns to (i) place representations of similar symbols in close proximity in a vector space, (ii) make use of such similarities to prove queries, (iii) induce logical rules, and (iv) use provided and induced logical rules for multi-hop reasoning. We demonstrate that this architecture outperforms ComplEx, a state-of-the-art neural link prediction model, on three out of four benchmark knowledge bases while at the same time inducing interpretable function-free first-order logic rules.

## 1  Introduction

Current state-of-the-art methods for automated Knowledge Base (KB) completion use neural link prediction models to learn distributed vector representations of symbols (*i.e.* subsymbolic representations) for scoring fact triples [1–7]. Such subsymbolic representations enable these models to generalize to unseen facts by encoding similarities: If the vector of the predicate symbol grandfatherOf is similar to the vector of the symbol grandpaOf, both predicates likely express a similar relation. Likewise, if the vector of the constant symbol LISA is similar to MAGGIE, similar relations likely hold for both constants (*e.g.* they live in the same city, have the same parents etc.).

This simple form of reasoning based on similarities is remarkably effective for automatically completing large KBs. However, in practice it is often important to capture more complex reasoning patterns that involve several inference steps. For example, if ABE is the father of HOMER and HOMER is a parent of BART, we would like to infer that ABE is a grandfather of BART. Such transitive reasoning is inherently hard for neural link prediction models as they only learn to score facts locally. In contrast, symbolic theorem provers like Prolog [8] enable exactly this type of multi-hop reasoning. Furthermore, Inductive Logic Programming (ILP) [9] builds upon such provers to learn interpretable rules from data and to exploit them for reasoning in KBs. However, symbolic provers lack the ability to learn subsymbolic representations and similarities between them from large KBs, which limits their ability to generalize to queries with similar but not identical symbols.

While the connection between logic and machine learning has been addressed by statistical relational learning approaches, these models traditionally do not support reasoning with subsymbolic representations (*e.g.* [10]), and when using subsymbolic representations they are not trained end-to-end from training data (*e.g.* [11–13]). Neural multi-hop reasoning models [14–18] address the aforementioned limitations to some extent by encoding reasoning chains in a vector space or by iteratively refining subsymbolic representations of a question before comparison with answers. In many ways, these models operate like basic theorem provers, but they lack two of their most crucial ingredients:

interpretability and straightforward ways of incorporating domain-specific knowledge in form of rules.

Our approach to this problem is inspired by recent neural network architectures like Neural Turing Machines [19], Memory Networks [20], Neural Stacks/Queues [21, 22], Neural Programmer [23], Neural Programmer-Interpreters [24], Hierarchical Attentive Memory [25] and the Differentiable Forth Interpreter [26]. These architectures replace discrete algorithms and data structures by end-to-end differentiable counterparts that operate on real-valued vectors. At the heart of our approach is the idea to translate this concept to basic symbolic theorem provers, and hence combine their advantages (multi-hop reasoning, interpretability, easy integration of domain knowledge) with the ability to reason with vector representations of predicates and constants. Specifically, we keep variable binding symbolic but compare symbols using their subsymbolic vector representations.

Concretely, we introduce Neural Theorem Provers (NTPs): End-to-end differentiable provers for basic theorems formulated as queries to a KB. We use Prolog's backward chaining algorithm as a recipe for recursively constructing neural networks that are capable of proving queries to a KB using subsymbolic representations. The success score of such proofs is differentiable with respect to vector representations of symbols, which enables us to learn such representations for predicates and constants in ground atoms, as well as parameters of function-free first-order logic rules of predefined structure. By doing so, NTPs learn to place representations of similar symbols in close proximity in a vector space and to induce rules given prior assumptions about the structure of logical relationships in a KB such as transitivity. Furthermore, NTPs can seamlessly reason with provided domain-specific rules. As NTPs operate on distributed representations of symbols, a single hand-crafted rule can be leveraged for many proofs of queries with symbols that have a similar representation. Finally, NTPs demonstrate a high degree of interpretability as they induce latent rules that we can decode to human-readable symbolic rules.

Our contributions are threefold: (i) We present the construction of NTPs inspired by Prolog's backward chaining algorithm and a differentiable unification operation using subsymbolic representations, (ii) we propose optimizations to this architecture by joint training with a neural link prediction model, batch proving, and approximate gradient calculation, and (iii) we experimentally show that NTPs can learn representations of symbols and function-free first-order rules of predefined structure, enabling them to learn to perform multi-hop reasoning on benchmark KBs and to outperform ComplEx [7], a state-of-the-art neural link prediction model, on three out of four KBs.

## 2   Background

In this section, we briefly introduce the syntax of KBs that we use in the remainder of the paper. We refer the reader to [27, 28] for a more in-depth introduction. An *atom* consists of a *predicate* symbol and a list of terms. We will use lowercase names to refer to predicate and constant symbols (*e.g.* fatherOf and BART), and uppercase names for variables (*e.g.* X, Y, Z). As we only consider function-free first-order logic rules, a *term* can only be a constant or a variable. For instance, [grandfatherOf, Q, BART] is an atom with the predicate grandfatherOf, and two terms, the variable Q and the constant BART. We consider *rules* of the form H :– $\mathbb{B}$, where the body $\mathbb{B}$ is a possibly empty conjunction of atoms represented as a list, and the head H is an atom. We call a rule with no free variables a ground rule. All variables are universally quantified. We call a ground rule with an empty body a *fact*. A *substitution set* $\psi = \{X_1/t_1, \ldots, X_N/t_N\}$ is an assignment of variable symbols $X_i$ to terms $t_i$, and applying substitutions to an atom replaces all occurrences of variables $X_i$ by their respective term $t_i$.

Given a query (also called goal) such as [grandfatherOf, Q, BART], we can use Prolog's backward chaining algorithm to find substitutions for Q [8] (see appendix A for pseudocode). On a high level, backward chaining is based on two functions called OR and AND. OR iterates through all rules (including rules with an empty body, *i.e.*, facts) in a KB and unifies the goal with the respective rule head, thereby updating a substitution set. It is called OR since any successful proof suffices (disjunction). If unification succeeds, OR calls AND to prove all atoms (subgoals) in the body of the rule. To prove subgoals of a rule body, AND first applies substitutions to the first atom that is then proven by again calling OR, before proving the remaining subgoals by recursively calling AND. This function is called AND as all atoms in the body need to be proven together (conjunction). As an example, a rule such as [grandfatherOf, X, Y] :– [[fatherOf, X, Z], [parentOf, Z, Y]] is used

in OR for translating a goal like $[\texttt{grandfatherOf}, \text{Q}, \text{BART}]$ into subgoals $[\texttt{fatherOf}, \text{Q}, \text{Z}]$ and $[\texttt{parentOf}, \text{Z}, \text{BART}]$ that are subsequently proven by AND.[1]

## 3  Differentiable Prover

In the following, we describe the recursive construction of NTPs – neural networks for end-to-end differentiable proving that allow us to calculate the gradient of proof successes with respect to vector representations of symbols. We define the construction of NTPs in terms of *modules* similar to dynamic neural module networks [29]. Each module takes as inputs *discrete objects* (atoms and rules) and a *proof state*, and returns a list of new proof states (see Figure 1 for a graphical representation).

A proof state $S = (\psi, \rho)$ is a tuple consisting of the substitution set $\psi$ constructed in the proof so far and a neural network $\rho$ that outputs a real-valued success score of a (partial) proof. While discrete objects and the substitution set are only used during construction of the neural network, once the network is constructed a continuous proof success score can be calculated for many different goals at training and test time. To summarize, modules are instantiated by discrete objects and the substitution set. They construct a neural network representing the (partial) proof success score and recursively instantiate submodules to continue the proof.

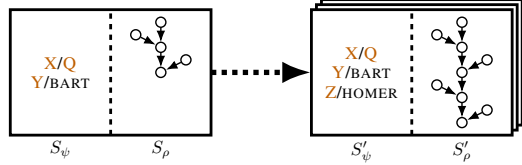

Figure 1: A module is mapping an upstream proof state (left) to a list of new proof states (right), thereby extending the substitution set $S_\psi$ and adding nodes to the computation graph of the neural network $S_\rho$ representing the proof success.

The shared signature of modules is $\mathcal{D} \times \mathcal{S} \to \mathcal{S}^N$ where $\mathcal{D}$ is a domain that controls the construction of the network, $\mathcal{S}$ is the domain of proof states, and $N$ is the number of output proof states. Furthermore, let $S_\psi$ denote the substitution set of the proof state $S$ and let $S_\rho$ denote the neural network for calculating the proof success.

We use pseudocode in style of a functional programming language to define the behavior of modules and auxiliary functions. Particularly, we are making use of pattern matching to check for properties of arguments passed to a module. We denote sets by Euler script letters (*e.g.* $\mathcal{E}$), lists by small capital letters (*e.g.* E), lists of lists by blackboard bold letters (*e.g.* $\mathbb{E}$) and we use : to refer to prepending an element to a list (*e.g.* $e : \text{E}$ or $\text{E} : \mathbb{E}$). While an atom is a list of a predicate symbol and terms, a rule can be seen as a list of atoms and thus a list of lists where the head of the list is the rule head.[2]

### 3.1  Unification Module

Unification of two atoms, *e.g.*, a goal that we want to prove and a rule head, is a central operation in backward chaining. Two non-variable symbols (predicates or constants) are checked for equality and the proof can be aborted if this check fails. However, we want to be able to apply rules even if symbols in the goal and head are not equal but similar in meaning (*e.g.* $\texttt{grandfatherOf}$ and $\texttt{grandpaOf}$) and thus replace symbolic comparison with a computation that measures the similarity of both symbols in a vector space.

The module $\texttt{unify}$ updates a substitution set and creates a neural network for comparing the vector representations of non-variable symbols in two sequences of terms. The signature of this module is $\mathcal{L} \times \mathcal{L} \times \mathcal{S} \to \mathcal{S}$ where $\mathcal{L}$ is the domain of lists of terms. $\texttt{unify}$ takes two atoms represented as lists of terms and an upstream proof state, and maps these to a new proof state (substitution set and proof success). To this end, $\texttt{unify}$ iterates through the list of terms of two atoms and compares their symbols. If one of the symbols is a variable, a substitution is added to the substitution set. Otherwise, the vector representations of the two non-variable symbols are compared using a Radial Basis Function (RBF) kernel [30] where $\mu$ is a hyperparameter that we set to $\frac{1}{\sqrt{2}}$ in our experiments. The following pseudocode implements $\texttt{unify}$. Note that "_" matches every argument and that the

order matters, *i.e.*, if arguments match a line, subsequent lines are not evaluated.

1. $\texttt{unify}_{\boldsymbol{\theta}}([\,],[\,],S) = S$
2. $\texttt{unify}_{\boldsymbol{\theta}}([\,],\_,\_) = \texttt{FAIL}$
3. $\texttt{unify}_{\boldsymbol{\theta}}(\_,[\,],\_) = \texttt{FAIL}$
4. $\texttt{unify}_{\boldsymbol{\theta}}(h:\text{H}, g:\text{G}, S) = \texttt{unify}_{\boldsymbol{\theta}}(\text{H},\text{G},S') = (S'_{\psi}, S'_{\rho})$   where

$$
S'_{\psi} = \left\{
\begin{array}{ll}
S_{\psi} \cup \{h/g\} & \text{if } h \in \mathcal{V} \\
S_{\psi} \cup \{g/h\} & \text{if } g \in \mathcal{V}, h \notin \mathcal{V} \\
S_{\psi} & \text{otherwise}
\end{array}
\right\}, \quad
S'_{\rho} = \min\left( S_{\rho}, \left\{
\begin{array}{ll}
\exp\left(\frac{-\|\boldsymbol{\theta}_{h:} - \boldsymbol{\theta}_{g:}\|_2}{2\mu^2}\right) & \text{if } h,g \notin \mathcal{V} \\
1 & \text{otherwise}
\end{array}
\right\} \right)
$$

Here, $S'$ refers to the new proof state, $\mathcal{V}$ refers to the set of variable symbols, $h/g$ is a substitution from the variable symbol $h$ to the symbol $g$, and $\boldsymbol{\theta}_{g:}$ denotes the embedding lookup of the non-variable symbol with index $g$. $\texttt{unify}$ is parameterized by an embedding matrix $\boldsymbol{\theta} \in \mathbb{R}^{|\mathcal{Z}| \times k}$ where $\mathcal{Z}$ is the set of non-variables symbols and $k$ is the dimension of vector representations of symbols. Furthermore, $\texttt{FAIL}$ represents a unification failure due to mismatching arity of two atoms. Once a failure is reached, we abort the creation of the neural network for this branch of proving. In addition, we constrain proofs to be cycle-free by checking whether a variable is already bound. Note that this is a simple heuristic that prohibits applying the same non-ground rule twice. There are more sophisticated ways for finding and avoiding cycles in a proof graph such that the same rule can still be applied multiple times (*e.g.* [31]), but we leave this for future work.

**Example**  Assume that we are unifying two atoms $[\texttt{grandpaOf}, \text{ABE}, \text{BART}]$ and $[s, \text{Q}, i]$ given an upstream proof state $S = (\varnothing, \rho)$ where the latter input atom has placeholders for a predicate $s$ and a constant $i$, and the neural network $\rho$ would output $0.7$ when evaluated. Furthermore, assume $\texttt{grandpaOf}$, ABE and BART represent the indices of the respective symbols in a global symbol vocabulary. Then, the new proof state constructed by $\texttt{unify}$ is:

$$
\texttt{unify}_{\boldsymbol{\theta}}([\texttt{grandpaOf}, \text{ABE}, \text{BART}], [s, \text{Q}, i], (\varnothing, \rho)) = (S'_{\psi}, S'_{\rho}) =
$$
$$
\left( \{\text{Q}/\text{ABE}\}, \min\left( \rho, \exp(-\|\boldsymbol{\theta}_{\texttt{grandpaOf}:} - \boldsymbol{\theta}_{s:}\|_2), \exp(-\|\boldsymbol{\theta}_{\text{BART}:} - \boldsymbol{\theta}_{i:}\|_2) \right) \right)
$$

Thus, the output score of the neural network $S'_{\rho}$ will be high if the subsymbolic representation of the input $s$ is close to $\texttt{grandpaOf}$ and the input $i$ is close to BART. However, the score cannot be higher than $0.7$ due to the upstream proof success score in the forward pass of the neural network $\rho$. Note that in addition to extending the neural networks $\rho$ to $S'_{\rho}$, this module also outputs a substitution set $\{\text{Q}/\text{ABE}\}$ at graph creation time that will be used to instantiate submodules.

### 3.2 OR Module

Based on $\texttt{unify}$, we now define the $\texttt{or}$ module which attempts to apply rules in a KB. The signature of $\texttt{or}$ is $\mathcal{L} \times \mathbb{N} \times \mathcal{S} \to \mathcal{S}^N$ where $\mathcal{L}$ is the domain of goal atoms and $\mathbb{N}$ is the domain of integers used for specifying the maximum proof depth of the neural network. Furthermore, $N$ is the number of possible output proof states for a goal of a given structure and a provided KB.[3] We implement $\texttt{or}$ as

1. $\texttt{or}_{\boldsymbol{\theta}}^{\mathfrak{K}}(\text{G}, d, S) = [S' \mid S' \in \texttt{and}_{\boldsymbol{\theta}}^{\mathfrak{K}}(\mathbb{B}, d, \texttt{unify}_{\boldsymbol{\theta}}(\text{H}, \text{G}, S)) \text{ for } \text{H} :\!\!- \mathbb{B} \in \mathfrak{K}]$

where $\text{H} :\!\!- \mathbb{B}$ denotes a rule in a given KB $\mathfrak{K}$ with a head atom H and a list of body atoms $\mathbb{B}$. In contrast to the symbolic OR method, the $\texttt{or}$ module is able to use the $\texttt{grandfatherOf}$ rule above for a query involving $\texttt{grandpaOf}$ provided that the subsymbolic representations of both predicates are similar as measured by the RBF kernel in the $\texttt{unify}$ module.

**Example**  For a goal $[s, \text{Q}, i]$, $\texttt{or}$ would instantiate an $\texttt{and}$ submodule based on the rule $[\texttt{grandfatherOf}, \text{X}, \text{Y}] :\!\!- [[\texttt{fatherOf}, \text{X}, \text{Z}], [\texttt{parentOf}, \text{Z}, \text{Y}]]$ as follows

$$
\texttt{or}_{\boldsymbol{\theta}}^{\mathfrak{K}}([s, \text{Q}, i], d, S) = [S' | S' \in \texttt{and}_{\boldsymbol{\theta}}^{\mathfrak{K}}([[\texttt{fatherOf}, \text{X}, \text{Z}], [\texttt{parentOf}, \text{Z}, \text{Y}]], d, \underbrace{(\{\text{X}/\text{Q}, \text{Y}/i\}, \hat{S}_{\rho})}_{\text{result of } \texttt{unify}}), \ldots]
$$

### 3.3 AND Module

For implementing `and` we first define an auxiliary function called substitute which applies substitutions to variables in an atom if possible. This is realized via

1. $\text{substitute}([\,],\_) = [\,]$

2. $\text{substitute}(g : \text{G}, \psi) = \left\{ \begin{array}{ll} x & \text{if } g/x \in \psi \\ g & \text{otherwise} \end{array} \right\} : \text{substitute}(\text{G}, \psi)$

For example, $\text{substitute}([\texttt{fatherOf}, \text{X}, \text{Z}], \{\text{X}/\text{Q}, \text{Y}/i\})$ results in $[\texttt{fatherOf}, \text{Q}, \text{Z}]$.

The signature of `and` is $\mathcal{L} \times \mathbb{N} \times \mathcal{S} \to \mathcal{S}^N$ where $\mathcal{L}$ is the domain of lists of atoms and $N$ is the number of possible output proof states for a list of atoms with a known structure and a provided KB. This module is implemented as

1. $\text{and}_{\boldsymbol{\theta}}^{\mathfrak{K}}(\_, \_, \texttt{FAIL}) = \texttt{FAIL}$

2. $\text{and}_{\boldsymbol{\theta}}^{\mathfrak{K}}(\_, 0, \_) = \texttt{FAIL}$

3. $\text{and}_{\boldsymbol{\theta}}^{\mathfrak{K}}([\,], \_, S) = S$

4. $\text{and}_{\boldsymbol{\theta}}^{\mathfrak{K}}(\text{G} : \mathbb{G}, d, S) = [S'' \mid S'' \in \text{and}_{\boldsymbol{\theta}}^{\mathfrak{K}}(\mathbb{G}, d, S') \text{ for } S' \in \text{or}_{\boldsymbol{\theta}}^{\mathfrak{K}}(\text{substitute}(\text{G}, S_\psi), d - 1, S)]$

where the first two lines define the failure of a proof, either because of an upstream unification failure that has been passed from the `or` module (line 1), or because the maximum proof depth has been reached (line 2). Line 3 specifies a proof success, *i.e.*, the list of subgoals is empty before the maximum proof depth has been reached. Lastly, line 4 defines the recursion: The first subgoal G is proven by instantiating an `or` module after substitutions are applied, and every resulting proof state $S'$ is used for proving the remaining subgoals $\mathbb{G}$ by again instantiating `and` modules.

**Example** Continuing the example from Section 3.2, the `and` module would instantiate submodules as follows:

$$\text{and}_{\boldsymbol{\theta}}^{\mathfrak{K}}([[\texttt{fatherOf}, \text{X}, \text{Z}], [\texttt{parentOf}, \text{Z}, \text{Y}]], d, \underbrace{(\{\text{X}/\text{Q}, \text{Y}/i\}, \hat{S}_\rho)}_{\text{result of unify in or}}) =$$

$$[S'' | S'' \in \text{and}_{\boldsymbol{\theta}}^{\mathfrak{K}}([[\texttt{parentOf}, \text{Z}, \text{Y}]], d, S') \text{ for } S' \in \text{or}_{\boldsymbol{\theta}}^{\mathfrak{K}}(\underbrace{[\texttt{fatherOf}, \text{Q}, \text{Z}]}_{\text{result of substitute}}, d - 1, \underbrace{(\{\text{X}/\text{Q}, \text{Y}/i\}, \hat{S}_\rho)}_{\text{result of unify in or}})]$$

### 3.4 Proof Aggregation

Finally, we define the overall success score of proving a goal G using a KB $\mathfrak{K}$ with parameters $\boldsymbol{\theta}$ as

$$\text{ntp}_{\boldsymbol{\theta}}^{\mathfrak{K}}(\text{G}, d) = \underset{\substack{S \in \text{or}_{\boldsymbol{\theta}}^{\mathfrak{K}}(\text{G}, d, (\varnothing, 1)) \\ S \neq \texttt{FAIL}}}{\arg\max} S_\rho$$

where $d$ is a predefined maximum proof depth and the initial proof state is set to an empty substitution set and a proof success score of 1.

**Example** Figure 2 illustrates an examplary NTP computation graph constructed for a toy KB. Note that such an NTP is constructed once before training, and can then be used for proving goals of the structure $[s, i, j]$ at training and test time where $s$ is the index of an input predicate, and $i$ and $j$ are indices of input constants. Final proof states which are used in proof aggregation are underlined.

### 3.5 Neural Inductive Logic Programming

We can use NTPs for ILP by gradient descent instead of a combinatorial search over the space of rules as, for example, done by the First Order Inductive Learner (FOIL) [32]. Specifically, we are using the concept of learning from entailment [9] to induce rules that let us prove known ground atoms, but that do not give high proof success scores to sampled unknown ground atoms.

Let $\boldsymbol{\theta}_{r:}, \boldsymbol{\theta}_{s:}, \boldsymbol{\theta}_{t:} \in \mathbb{R}^k$ be representations of some unknown predicates with indices $r$, $s$ and $t$ respectively. The prior knowledge of a transitivity between three unknown predicates can be specified via

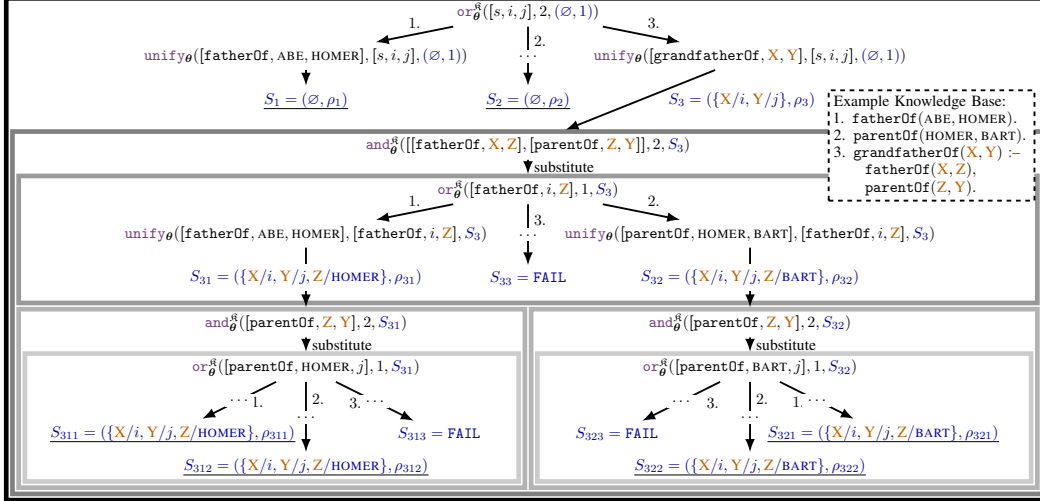

Figure 2: Exemplary construction of an NTP computation graph for a toy knowledge base. Indices on arrows correspond to application of the respective KB rule. Proof states (blue) are subscripted with the sequence of indices of the rules that were applied. Underlined proof states are aggregated to obtain the final proof success. Boxes visualize instantiations of modules (omitted for unify). The proofs $S_{33}$, $S_{313}$ and $S_{323}$ fail due to cycle-detection (the same rule cannot be applied twice).

$r(\mathrm{X}, \mathrm{Y}) :\!\!- s(\mathrm{X}, \mathrm{Z}), t(\mathrm{Z}, \mathrm{Y})$. We call this a *parameterized rule* as the corresponding predicates are unknown and their representations are learned from data. Such a rule can be used for proofs at training and test time in the same way as any other given rule. During training, the predicate representations of parameterized rules are optimized jointly with all other subsymbolic representations. Thus, the model can adapt parameterized rules such that proofs for known facts succeed while proofs for sampled unknown ground atoms fail, thereby inducing rules of predefined structures like the one above. Inspired by [33], we use rule templates for conveniently defining the structure of multiple parameterized rules by specifying the number of parameterized rules that should be instantiated for a given rule structure (see appendix E for examples). For inspection after training, we decode a parameterized rule by searching for the closest representations of known predicates. In addition, we provide users with a rule confidence by taking the minimum similarity between unknown and decoded predicate representations using the RBF kernel in `unify`. This confidence score is an upper bound on the proof success score that can be achieved when the induced rule is used in proofs.

## 4 Optimization

In this section, we present the basic training loss that we use for NTPs, a training loss where a neural link prediction models is used as auxiliary task, as well as various computational optimizations.

### 4.1 Training Objective

Let $\mathcal{K}$ be the set of known facts in a given KB. Usually, we do not observe negative facts and thus resort to sampling corrupted ground atoms as done in previous work [34]. Specifically, for every $[s, i, j] \in \mathcal{K}$ we obtain corrupted ground atoms $[s, \hat{i}, j], [s, i, \hat{j}], [s, \tilde{i}, \tilde{j}] \notin \mathcal{K}$ by sampling $\hat{i}, \hat{j}, \tilde{i}$ and $\tilde{j}$ from the set of constants. These corrupted ground atoms are resampled in every iteration of training, and we denote the set of known and corrupted ground atoms together with their target score (1.0 for known ground atoms and 0.0 for corrupted ones) as $\mathcal{T}$. We use the negative log-likelihood of the proof success score as loss function for an NTP with parameters $\boldsymbol{\theta}$ and a given KB $\mathfrak{K}$

$$\mathcal{L}_{\mathtt{ntp}_{\boldsymbol{\theta}}^{\mathfrak{K}}} = \sum_{([s,i,j],y)\,\in\,\mathcal{T}} -y \log(\mathtt{ntp}_{\boldsymbol{\theta}}^{\mathfrak{K}}([s,i,j],d)_{\rho}) - (1-y)\log(1 - \mathtt{ntp}_{\boldsymbol{\theta}}^{\mathfrak{K}}([s,i,j],d)_{\rho})$$

where $[s, i, j]$ is a training ground atom and $y$ its target proof success score. Note that since in our application all training facts are ground atoms, we only make use of the proof success score $\rho$ and not

the substitution list of the resulting proof state. We can prove known facts trivially by a unification with themselves, resulting in no parameter updates during training and hence no generalization. Therefore, during training we are masking the calculation of the unification success of a known ground atom that we want to prove. Specifically, we set the unification score to 0 to temporarily hide that training fact and assume it can be proven from other facts and rules in the KB.

## 4.2 Neural Link Prediction as Auxiliary Loss

At the beginning of training all subsymbolic representations are initialized randomly. When unifying a goal with all facts in a KB we consequently get very noisy success scores in early stages of training. Moreover, as only the maximum success score will result in gradient updates for the respective subsymbolic representations along the maximum proof path, it can take a long time until NTPs learn to place similar symbols close to each other in the vector space and to make effective use of rules.

To speed up learning subsymbolic representations, we train NTPs jointly with ComplEx [7] (Appendix B). ComplEx and the NTP share the same subsymbolic representations, which is feasible as the RBF kernel in `unify` is also defined for complex vectors. While the NTP is responsible for multi-hop reasoning, the neural link prediction model learns to score ground atoms locally. At test time, only the NTP is used for predictions. Thus, the training loss for ComplEx can be seen as an auxiliary loss for the subsymbolic representations learned by the NTP. We term the resulting model NTP$\lambda$. Based on the loss in Section 4.1, the joint training loss is defined as

$$\mathcal{L}_{\mathtt{ntp}\lambda_{\boldsymbol{\theta}}^{\mathfrak{K}}} = \mathcal{L}_{\mathtt{ntp}_{\boldsymbol{\theta}}^{\mathfrak{K}}} + \sum_{([s,i,j],y) \,\in\, \mathcal{T}} -y \log(\mathtt{complex}_{\boldsymbol{\theta}}(s,i,j)) - (1-y) \log(1 - \mathtt{complex}_{\boldsymbol{\theta}}(s,i,j))$$

where $[s,i,j]$ is a training atom and $y$ its ground truth target.

## 4.3 Computational Optimizations

NTPs as described above suffer from severe computational limitations since the neural network is representing all possible proofs up to some predefined depth. In contrast to symbolic backward chaining where a proof can be aborted as soon as unification fails, in differentiable proving we only get a unification failure for atoms whose arity does not match or when we detect cyclic rule application. We propose two optimizations to speed up NTPs in the Appendix. First, we make use of modern GPUs by batch processing many proofs in parallel (Appendix C). Second, we exploit the sparseness of gradients caused by the min and max operations used in the unification and proof aggregation respectively to derive a heuristic for a truncated forward and backward pass that drastically reduces the number of proofs that have to be considered for calculating gradients (Appendix D).

# 5 Experiments

Consistent with previous work, we carry out experiments on four benchmark KBs and compare ComplEx with the NTP and NTP$\lambda$ in terms of area under the Precision-Recall-curve (AUC-PR) on the Countries KB, and Mean Reciprocal Rank (MRR) and HITS@$m$ [34] on the other KBs described below. Training details, including hyperparameters and rule templates, can be found in Appendix E.

**Countries**   The Countries KB is a dataset introduced by [35] for testing reasoning capabilities of neural link prediction models. It consists of 244 countries, 5 regions (*e.g.* EUROPE), 23 subregions (*e.g.* WESTERN EUROPE, NORTHERN AMERICA), and 1158 facts about the neighborhood of countries, and the location of countries and subregions. We follow [36] and split countries randomly into a training set of 204 countries (train), a development set of 20 countries (dev), and a test set of 20 countries (test), such that every dev and test country has at least one neighbor in the training set. Subsequently, three different task datasets are created. For all tasks, the goal is to predict $\mathtt{locatedIn}(c, r)$ for every test country $c$ and all five regions $r$, but the access to training atoms in the KB varies.
**S1:** All ground atoms $\mathtt{locatedIn}(c, r)$ where $c$ is a test country and $r$ is a region are removed from the KB. Since information about the subregion of test countries is still contained in the KB, this task can be solved by using the transitivity rule $\mathtt{locatedIn}(\mathrm{X}, \mathrm{Y}) :\!-\, \mathtt{locatedIn}(\mathrm{X}, \mathrm{Z}), \mathtt{locatedIn}(\mathrm{Z}, \mathrm{Y})$.
**S2:** In addition to **S1**, all ground atoms $\mathtt{locatedIn}(c, s)$ are removed where $c$ is a test country and $s$

Table 1: AUC-PR results on Countries and MRR and HITS@$m$ on Kinship, Nations, and UMLS.

| Corpus | | Metric | Model | | | Examples of induced rules and their confidence |
|---|---|---|---|---|---|---|
| | | | **ComplEx** | **NTP** | **NTPλ** | |
| Countries | S1 | AUC-PR | $99.37 \pm 0.4$ | $90.83 \pm 15.4$ | $\mathbf{100.00 \pm 0.0}$ | 0.90 `locatedIn(X,Y) :- locatedIn(X,Z), locatedIn(Z,Y).` |
| | S2 | AUC-PR | $87.95 \pm 2.8$ | $87.40 \pm 11.7$ | $\mathbf{93.04 \pm 0.4}$ | 0.63 `locatedIn(X,Y) :- neighborOf(X,Z), locatedIn(Z,Y).` |
| | S3 | AUC-PR | $48.44 \pm 6.3$ | $56.68 \pm 17.6$ | $\mathbf{77.26 \pm 17.0}$ | 0.32 `locatedIn(X,Y) :-` |
| | | | | | | `neighborOf(X,Z), neighborOf(Z,W), locatedIn(W,Y).` |
| Kinship | | MRR | **0.81** | 0.60 | 0.80 | 0.98 `term15(X,Y) :- term5(Y,X)` |
| | | HITS@1 | 0.70 | 0.48 | **0.76** | 0.97 `term18(X,Y) :- term18(Y,X)` |
| | | HITS@3 | **0.89** | 0.70 | 0.82 | 0.86 `term4(X,Y) :- term4(Y,X)` |
| | | HITS@10 | **0.98** | 0.78 | 0.89 | 0.73 `term12(X,Y) :- term10(X, Z), term12(Z, Y).` |
| Nations | | MRR | **0.75** | **0.75** | 0.74 | 0.68 `blockpositionindex(X,Y) :- blockpositionindex(Y,X).` |
| | | HITS@1 | **0.62** | **0.62** | 0.59 | 0.46 `expeldiplomats(X,Y) :- negativebehavior(X,Y).` |
| | | HITS@3 | 0.84 | 0.86 | **0.89** | 0.38 `negativecomm(X,Y) :- commonbloc0(X,Y).` |
| | | HITS@10 | **0.99** | **0.99** | **0.99** | 0.38 `intergovorgs3(X,Y) :- intergovorgs(Y,X).` |
| UMLS | | MRR | 0.89 | 0.88 | **0.93** | 0.88 `interacts_with(X,Y) :-` |
| | | HITS@1 | 0.82 | 0.82 | **0.87** | `interacts_with(X,Z), interacts_with(Z,Y).` |
| | | HITS@3 | 0.96 | 0.92 | **0.98** | 0.77 `isa(X,Y) :- isa(X,Z), isa(Z,Y).` |
| | | HITS@10 | **1.00** | 0.97 | **1.00** | 0.71 `derivative_of(X,Y) :-` |
| | | | | | | `derivative_of(X,Z), derivative_of(Z,Y).` |

is a subregion. The location of test countries needs to be inferred from the location of its neighboring countries: $\texttt{locatedIn}(X, Y) :- \texttt{neighborOf}(X, Z), \texttt{locatedIn}(Z, Y)$. This task is more difficult than **S1**, as neighboring countries might not be in the same region, so the rule above will not always hold.

**S3:** In addition to **S2**, all ground atoms $\texttt{locatedIn}(c, r)$ where $r$ is a region and $c$ is a training country that has a test or dev country as a neighbor are also removed. The location of test countries can for instance be inferred using the three-hop rule $\texttt{locatedIn}(X, Y) :- \texttt{neighborOf}(X, Z), \texttt{neighborOf}(Z, W), \texttt{locatedIn}(W, Y)$.

**Kinship, Nations & UMLS** We use the Nations, Alyawarra kinship (Kinship) and Unified Medical Language System (UMLS) KBs from [10]. We left out the Animals dataset as it only contains unary predicates and can thus not be used for evaluating multi-hop reasoning. Nations contains 56 binary predicates, 111 unary predicates, 14 constants and 2565 true facts, Kinship contains 26 predicates, 104 constants and 10686 true facts, and UMLS contains 49 predicates, 135 constants and 6529 true facts. Since our baseline ComplEx cannot deal with unary predicates, we remove unary atoms from Nations. We split every KB into 80% training facts, 10% development facts and 10% test facts. For evaluation, we take a test fact and corrupt its first and second argument in all possible ways such that the corrupted fact is not in the original KB. Subsequently, we predict a ranking of every test fact and its corruptions to calculate MRR and HITS@$m$.

# 6   Results and Discussion

Results for the different model variants on the benchmark KBs are shown in Table 1. Another method for inducing rules in a differentiable way for automated KB completion has been introduced recently by [37] and our evaluation setup is equivalent to their Protocol II. However, our neural link prediction baseline, ComplEx, already achieves much higher HITS@10 results (1.00 vs. 0.70 on UMLS and 0.98 vs. 0.73 on Kinship). We thus focus on the comparison of NTPs with ComplEx.

First, we note that vanilla NTPs alone do not work particularly well compared to ComplEx. They only outperform ComplEx on Countries S3 and Nations, but not on Kinship or UMLS. This demonstrates the difficulty of learning subsymbolic representations in a differentiable prover from unification alone, and the need for auxiliary losses. The NTPλ with ComplEx as auxiliary loss outperforms the other models in the majority of tasks. The difference in AUC-PR between ComplEx and NTPλ is significant for all Countries tasks ($p < 0.0001$).

A major advantage of NTPs is that we can inspect induced rules which provide us with an interpretable representation of what the model has learned. The right column in Table 1 shows examples of induced rules by NTPλ (note that predicates on Kinship are anonymized). For Countries, the NTP recovered those rules that are needed for solving the three different tasks. On UMLS, the NTP induced transitivity rules. Those relationships are particularly hard to encode by neural link prediction models like ComplEx, as they are optimized to locally predict the score of a fact.

# 7 Related Work

Combining neural and symbolic approaches to relational learning and reasoning has a long tradition and let to various proposed architectures over the past decades (see [38] for a review). Early proposals for neural-symbolic networks are limited to *propositional rules* (*e.g.*, EBL-ANN [39], KBANN [40] and C-IL$^2$P [41]). Other neural-symbolic approaches focus on first-order inference, but do not learn subsymbolic vector representations from training facts in a KB (*e.g.*, SHRUTI [42], Neural Prolog [43], CLIP++ [44], Lifted Relational Neural Networks [45], and TensorLog [46]). Logic Tensor Networks [47] are in spirit similar to NTPs, but need to fully ground first-order logic rules. However, they support function terms, whereas NTPs currently only support function-free terms.

Recent question-answering architectures such as [15, 17, 18] translate query representations implicitly in a vector space without explicit rule representations and can thus not easily incorporate domain-specific knowledge. In addition, NTPs are related to random walk [48, 49, 11, 12] and path encoding models [14, 16]. However, instead of aggregating paths from random walks or encoding paths to predict a target predicate, reasoning steps in NTPs are explicit and only unification uses subsymbolic representations. This allows us to induce interpretable rules, as well as to incorporate prior knowledge either in the form of rules or in the form of rule templates which define the structure of logical relationships that we expect to hold in a KB. Another line of work [50–54] regularizes distributed representations via domain-specific rules, but these approaches do not learn such rules from data and only support a restricted subset of first-order logic. NTPs are constructed from Prolog's backward chaining and are thus related to Unification Neural Networks [55, 56]. However, NTPs operate on vector representations of symbols instead of scalar values, which are more expressive.

As NTPs can learn rules from data, they are related to ILP systems such as FOIL [32], Sherlock [57] and meta-interpretive learning of higher-order dyadic Datalog (Metagol) [58]. While these ILP systems operate on symbols and search over the discrete space of logical rules, NTPs work with subsymbolic representations and induce rules using gradient descent. Recently, [37] introduced a differentiable rule learning system based on TensorLog and a neural network controller similar to LSTMs [59]. Their method is more scalable than the NTPs introduced here. However, on UMLS and Kinship our baseline already achieved stronger generalization by learning subsymbolic representations. Still, scaling NTPs to larger KBs for competing with more scalable relational learning methods is an open problem that we seek to address in future work.

# 8 Conclusion and Future Work

We proposed an end-to-end differentiable prover for automated KB completion that operates on subsymbolic representations. To this end, we used Prolog's backward chaining algorithm as a recipe for recursively constructing neural networks that can be used to prove queries to a KB. Specifically, we introduced a differentiable unification operation between vector representations of symbols. The constructed neural network allowed us to compute the gradient of proof successes with respect to vector representations of symbols, and thus enabled us to train subsymbolic representations end-to-end from facts in a KB, and to induce function-free first-order logic rules using gradient descent. On benchmark KBs, our model outperformed ComplEx, a state-of-the-art neural link prediction model, on three out of four KBs while at the same time inducing interpretable rules.

To overcome the computational limitations of the end-to-end differentiable prover introduced in this paper, we want to investigate the use of hierarchical attention [25] and reinforcement learning methods such as Monte Carlo tree search [60, 61] that have been used for learning to play Go [62] and chemical synthesis planning [63]. In addition, we plan to support function terms in the future. Based on [64], we are furthermore interested in applying NTPs to automated proving of mathematical theorems, either in logical or natural language form, similar to recent approaches by [65] and [66].

## Acknowledgements

We thank Pasquale Minervini, Tim Dettmers, Matko Bosnjak, Johannes Welbl, Naoya Inoue, Kai Arulkumaran, and the anonymous reviewers for very helpful comments on drafts of this paper. This work has been supported by a Google PhD Fellowship in Natural Language Processing, an Allen Distinguished Investigator Award, and a Marie Curie Career Integration Award.

## Footnotes

[1] For clarity, we will sometimes omit lists when writing rules and atoms, *e.g.*, $\texttt{grandfatherOf}(\text{X}, \text{Y}) :\!\!- \texttt{fatherOf}(\text{X}, \text{Z}), \texttt{parentOf}(\text{Z}, \text{Y}).$

[2] For example, $[[\texttt{grandfatherOf}, \text{X}, \text{Y}], [\texttt{fatherOf}, \text{X}, \text{Z}], [\texttt{parentOf}, \text{Z}, \text{Y}]].$

[3]The creation of the neural network is dependent on the KB but also the structure of the goal. For instance, the goal $s(\text{Q}, i)$ would result in a different neural network, and hence a different number of output proof states, than $s(i, j)$.

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
