[Supplementary Material]

# Appendix

## A  Backward Chaining Pseudocode

Simplified pseudocode for symbolic backward chaining (cycle detection omitted for brevity, see [27, 31, 8] for details).

1. $\text{or}(\text{G}, S) = [S' \mid S' \in \text{and}(\mathbb{B}, \text{unify}(\text{H}, \text{G}, S)) \text{ for H} :\!\!- \mathbb{B} \in \mathfrak{K}]$

2. $\text{and}(\_, \texttt{FAIL}) = \texttt{FAIL}$
3. $\text{and}([\,], S) = S$
4. $\text{and}(\text{G} : \mathbb{G}, S) = [S'' \mid S'' \in \text{and}(\mathbb{G}, S') \text{ for } S' \in \text{or}(\text{substitute}(\text{G}, S), S)]$

5. $\text{unify}(\_, \_, \texttt{FAIL}) = \texttt{FAIL}$
6. $\text{unify}([\,], [\,], S) = S$
7. $\text{unify}([\,], \_, \_) = \texttt{FAIL}$
8. $\text{unify}(\_, [\,], \_) = \texttt{FAIL}$

9. $\text{unify}(h : \text{H}, g : \text{G}, S) = \text{unify}\left(\text{H}, \text{G}, \left\{ \begin{array}{ll} S \cup \{h/g\} & \text{if } h \in \mathcal{V} \\ S \cup \{g/h\} & \text{if } g \in \mathcal{V}, h \notin \mathcal{V} \\ S & \text{if } g = h \\ \texttt{FAIL} & \text{otherwise} \end{array} \right\} \right)$

10. $\text{substitute}([\,], \_) = [\,]$

11. $\text{substitute}(g : \text{G}, S) = \left\{ \begin{array}{ll} x & \text{if } g/x \in S \\ g & \text{otherwise} \end{array} \right\} : \text{substitute}(\text{G}, S)$

## B  ComplEx

ComplEx [7] is a state-of-the-art neural link prediction model that represents symbols as complex vectors. Let $\text{real}(\boldsymbol{\theta}_{i:})$ denote the real part and $\text{imag}(\boldsymbol{\theta}_{i:})$ the imaginary part of a complex vector $\boldsymbol{\theta}_{i:} \in \mathbb{C}^k$ representing the symbol with the $i$th index. The scoring function defined by ComplEx is

$$\texttt{complex}_{\boldsymbol{\theta}}(s, i, j) = \sigma\big( \text{real}(\boldsymbol{\theta}_{s:})^\top (\text{real}(\boldsymbol{\theta}_{i:}) \odot \text{real}(\boldsymbol{\theta}_{j:})) + \text{real}(\boldsymbol{\theta}_{s:})^\top (\text{imag}(\boldsymbol{\theta}_{i:}) \odot \text{imag}(\boldsymbol{\theta}_{j:})) +$$
$$\text{imag}(\boldsymbol{\theta}_{s:})^\top (\text{real}(\boldsymbol{\theta}_{i:}) \odot \text{imag}(\boldsymbol{\theta}_{j:})) - \text{imag}(\boldsymbol{\theta}_{s:})^\top (\text{imag}(\boldsymbol{\theta}_{i:}) \odot \text{real}(\boldsymbol{\theta}_{j:})))$$

where $\odot$ denotes the element-wise multiplication and $\sigma$ the sigmoid function. The benefit of ComplEx over other neural link prediction models such as RESCAL [1] or DistMult [5] is that by using complex vectors as subsymbolic representations it can capture symmetric as well as asymmetric relations.

## C  Batch Proving

Let $\boldsymbol{A} \in \mathbb{R}^{N \times k}$ be a matrix of $N$ subsymbolic representations that are to be unified with $M$ other representations $\boldsymbol{B} \in \mathbb{R}^{M \times k}$. We can adapt the unification module to calculate the unification success in a batched way using

$$\exp\left( - \sqrt{ \left( \left[ \begin{array}{c} \sum_{i=1}^{k} \boldsymbol{A}_{1i}^2 \\ \vdots \\ \sum_{i=1}^{k} \boldsymbol{A}_{Ni}^2 \end{array} \right] \mathbf{1}_M^\top \right) + \left( \mathbf{1}_N \left[ \begin{array}{c} \sum_{i=1}^{k} \boldsymbol{B}_{1i}^2 \\ \vdots \\ \sum_{i=1}^{k} \boldsymbol{B}_{Mi}^2 \end{array} \right]^\top \right) - 2\boldsymbol{A}\boldsymbol{B}^\top } \right) \in \mathbb{R}^{N \times M}$$

where $\mathbf{1}_N$ and $\mathbf{1}_M$ are vectors of $N$ and $M$ ones respectively, and the square root is taken element-wise. In practice, we partition the KB into rules that have the same structure and batch-unify goals with all rule heads per partition at the same time on a Graphics Processing Unit (GPU). Furthermore, substitution sets bind variables to vectors of symbol indices instead of single symbol indices, and min and max operations are taken per goal.

# D $K \max$ **Gradient Approximation**

NTPs allow us to calculate the gradient of proof success scores with respect to subsymbolic representations and rule parameters. While backpropagating through this large computation graph will give us the exact gradient, it is computationally infeasible for any reasonably-sized KB. Consider the parameterized rule $\boldsymbol{\theta}_{1:}(X, Y) := \boldsymbol{\theta}_{2:}(X, Z), \boldsymbol{\theta}_{3:}(Z, Y)$ and let us assume the given KB contains 1 000 facts with binary predicates. While X and Y will be bound to the respective representations in the goal, Z we will be substituted with every possible second argument of the 1 000 facts in the KB when proving the first atom in the body. Moreover, for each of these 1 000 substitutions, we will again need to compare with all facts in the KB when proving the second atom in the body of the rule, resulting in 1 000 000 proof success scores. However, note that since we use the max operator for aggregating the success of different proofs, only subsymbolic representations in one out of 1 000 000 proofs will receive gradients.

To overcome this computational limitation, we propose the following heuristic. We assume that when unifying the first atom with facts in the KB, it is unlikely for any unification successes below the top $K$ successes to attain the maximum proof success when unifying the remaining atoms in the body of a rule with facts in the KB. That is, after the unification of the first atom, we only keep the top $K$ substitutions and their success scores, and continue proving only with these. This means that all other partial proofs will not contribute to the forward pass at this stage, and consequently not receive any gradients on the backward pass of backpropagation. We term this the $K \max$ heuristic. Note that we cannot guarantee anymore that the gradient of the proof success is the exact gradient, but for a large enough $K$ we get a close approximation to the true gradient.

# E   **Training Details**

We use ADAM [67] with an initial learning rate of $0.001$ and a mini-batch size of 50 (10 known and 40 corrupted atoms) for optimization. We apply an $\ell_2$ regularization of $0.01$ to all model parameters, and clip gradient values at $[-1.0, 1.0]$. All subsymbolic representations and rule parameters are initialized using Xavier initialization [68]. We train all models for 100 epochs and repeat every experiment on the Countries corpus ten times. Statistical significance is tested using the independent $t$-test. All models are implemented in TensorFlow [69]. We use a maximum proof depth of $d = 2$ and add the following rule templates where the number in front of the rule template indicates how often a parameterized rule of the given structure will be instantiated. Note that a rule template such as $\#1(X, Y) := \#2(X, Z), \#2(Z, Y)$ specifies that the two predicate representations in the body are shared.

**Countries S1**
3 $\#1(X, Y) := \#1(Y, X)$.
3 $\#1(X, Y) := \#2(X, Z), \#2(Z, Y)$.

**Countries S2**
3 $\#1(X, Y) := \#1(Y, X)$.
3 $\#1(X, Y) := \#2(X, Z), \#2(Z, Y)$.
3 $\#1(X, Y) := \#2(X, Z), \#3(Z, Y)$.

**Countries S3**
3 $\#1(X, Y) := \#1(Y, X)$.
3 $\#1(X, Y) := \#2(X, Z), \#2(Z, Y)$.
3 $\#1(X, Y) := \#2(X, Z), \#3(Z, Y)$.
3 $\#1(X, Y) := \#2(X, Z), \#3(Z, W), \#4(W, Y)$.

**Kinship, Nations & UMLS**
20 $\#1(X, Y) := \#2(X, Y)$.
20 $\#1(X, Y) := \#2(Y, X)$.
20 $\#1(X, Y) := \#2(X, Z), \#3(Z, Y)$.