[Reviews · NeurIPS 2017]

Reviewer 1



The paper describes a unified approach to learning deep neural networks with a symbolic AI approach. Specifically, based on prolog's backward conditioning approach, the neural network is constructed recursively to prove facts in a Knowledge base. The authors develop differentiable operations for unification, AND and OR. Vector representations of symbols help in sub symbolic matching for symbols that are similar but not identical. Experiments are conducted on three KBs. Overall the paper is quite good. It unifies two very popular areas in AI/ML and therefore I would expect this to be significant and something that could potentially result in a lot of follow-up work. The writing is good, maybe the background is a little too short. The results seem to be on-par with ComplEx, but the interpretability of a the results is a plus in the proposed system. The experiments are not run on particularly large KBs, so is scalability an issue. From what I understand, the computational complexity of the proposed system could be quite high. One thing that was not so clear to me is, it turns out that NTP-lambda is often better than NTP. Since this uses, ComplEx, I was not so sure of its role in the learning. Specifically, are a subset of the parameters learned by ComplEx, if so which subset, and how much effect does NTP learning have in this combined model.

Reviewer 2



Summary of paper ---------------- The paper presents a novel class of models, termed Neural Theorem Provers (NTPs), for automated knowledge base completion and automated theorem proving, using a deep neural network architecture. The recursive construction of the network is inspired by the backward chaining algorithm typically employed in logic programming (i.e., using the basic operations unification, conjunction and disjunction). Instead of directly operating on symbols, the neural network is employed to learn subsymbolic vector representations of entities and predicates, which are then exploited for assessing the similarity of symbols. Since the proposed architecture is fully differentiable, knowledge base completion can be performed using gradient descent. Thus, following the fundamental philosophy of neural-symbolic systems, the paper aims at combining the advantages of symbolic reasoning with those of subsymbolic inference. The paper briefly describes and discusses possible optimizations of the initial, computationally demanding algorithm (i.e., parallel evaluation of proofs and a truncation heuristic to speed up the gradient calculation), and proposes an interesting and apparently very effective training/regularization procedure based on the joint training of NTP together with a state-of-the-art neural link prediction method called ComplEx. Finally, it is shown that the performance of the proposed method on four benchmark knowledge base completion tasks is competitive against ComplEx. Also, it is argued that in contrast to previous approaches, the proposed algorithm is highly interpretable since it is capable of inductive logic programming, i.e., inducing concrete symbolic, first-order logic rules from parameterized templates. The paper concludes with some reasonable and likely worthwile ideas for future investigation. Short summary of review ----------------------- This is a solid paper. It aims at solving an important open problem and proposes a creative and effective solution by combining the efforts and thus aggregating the advantages of two very active lines of research. Particularly the dedication to interpretability of the method is commendable. The manuscript is rigorous and at the same time easy to follow. While there are a few issues concerning the quality and clarity, the overall article clearly has merit to be published at a venue like NIPS, and likely is of interest for several of its sub-communities. Quality ------- In general, the paper follows a high quality standard. While not being very theoretical in nature, the few technical details provided appear sound. While the experimental section impressively shows how NTPs excell at multi-hop reasoning and inducing interpretable first-order logic rules, it is not particularly successful at convincing the reader of the superiority of NTPs as compared to the state-of-the-art. This is because there is only a single competitor, ComplEx; employing a broader variety of competing algorithms (e.g. from the extensive pool of statistical relational learning methods) would certainly be commendable. However, this might be acceptable in the light of the contribution/focus of the paper being more on interpretability, rather than on significant performance increase. Another central issue with the paper is that while it prominently claims to have designed a system capable of 'theorem proving', the reader is actually never exposed to any evidence that the architecture at hand has indeed the ability to prove mathematical theorems. While this lack is acknowledged in the last sentence of the paper, the fact that said phrase is used in the first sentence of the abstract and even the title may provoke suspicions of deliberatley exploiting its catchiness for marketing purposes. Related work is discussed in commendable detail, providing a solid overview of the field, and extensive references are provided. Clarity ------- Most of the paper is well-organized and written in a clear, concise manner, making it rather easy to follow. Figure 1 is very helpful in understanding the general NTP architecture and the chosen running example (i.e., grandfatherOf) is simple enough to enable quick and intuitive understanding and at the same time expressive enough to convey and illustrate the central ideas of the method (also, the subtle reference to popular culture increases the enjoyability of reading the manuscript). However, the structure of section 3 is not quite reasonable: The paragraph directly following the heading '3 Methods' is only refering to subsections 3.1-3.3 (i.e., the modular building blocks of the architecture), but not to subsections 3.4-3.7, which appear to have been placed there rather arbitrarily. Maybe splitting section 3 into two distinct sections would help (e.g. '3 Architecture' and '4 Training & Inference' or something). Also, it would in my view make more sense to put the section on regularization (i.e., 3.7) immediately after the one on training (i.e., 3.4) since these are highly related and the former builds upon the latter (this would also enhance reading flow since one wouldn't have to scroll up to recall the formula for L_ntp, and to identify that the ntp-term and the complex-term share the same structure). Furthermore, the very short background section only defines terminology and notation, but does not provide any details on the fundamental mechanisms of logic programming (e.g. the backward chaining algorithm) that are required to be able to follow the manuscript. Given the tight page limit at NIPS it is evident that extensive coverage of preliminaries is not feasible, but the paper should at least provide pointers to relevant literature for those not familiar with the matter. Some further, specific issues: - it is not described how exactly the artificial neural network is created/constructed; it is stated in section 3.1 that 'The module unify [...] creates a neural network [...].' (l. 105-106), the specifics of which are not quite obvious; it should be detailed in what sense the min operation and the RBF kernel correspond to a neural network, and how the recursive construction of the network is to be understood; a graphical representation may be helpful here - there appear to be inconsistencies regarding the employed notation of data structures: atoms seem to be represented as lists (of three terms), meaning that rule bodies should be represented as lists of lists; however, in the definition of the AND module (section 3.3), sets (denoted by Euler script) are used instead of lists of lists for the first argument, which is also contradicting the signature of AND ('L is the domain of lists of atoms', l. 153); furthermore, the :: operator doesn't make sense on sets; the quickest fix for this issue would be to just define Euler scripts to denote lists of lists instead of sets - in the example for the unification module (section 3.1), where did the exponentials from the RBF kernel in the proof success score go? - in Figure 1, it is not clear why S_33, S_313 and S_323 fail; the UNIFY module can only fail if the arity of the atoms doesn't match, which doesn't appear to be the case here; thus, the computation can only fail due to reaching the maximum proof depth; however, as I see it, in that case further paths beyond (for example) S_33 must be explored first, meaning that so the operation cannot fail at S_33 already; this should be clarified - it would be helpful to have an inference example, i.e. an example of how the success score of a given statement is calculated, given the computational tree/graph in Figure 1; some kind of visualization of this would be helpful as well Originality ----------- As is discussed in detail, the paper is built upon the ideas of extensive prior work, and its main contribution essentially boils down to effectively combining two previously independent lines of research: 1) neural-symbolic systems for reasoning in knowledge bases, e.g. as developed in the field of statistical relational learning, and 2) the lately emerging end-to-end differentiable neural network architectures aiming at mimicking some of the capabilities of modern computers. Thus, the paper does not really provide any fundamental novel insight, but rather takes inspiration from said fields and symbiotically combines them in a novel and seemingly effective manner, succeeding at exploiting the advantages of both classes of approaches. The first line of work, i.e. neural-symbolic systems, aims at combining the benefits of subsymbolic similarity-based reasoning (and thus generalization) ability of neural link prediction / statistical relational learning with the symbolic multi-hop reasoning ability of automated theorem provers / logic programming systems. However, previous neural-symbolic approaches are either a) not trainable end-to-end, b) not interpretable and/or c) not capable of incorporating prior domain-specific knowledge. This is where the second line of work, i.e. fully differentiable computing neural network architectures, comes into play, enabling NTPs to handle all the features a), b) and c). Another contribution is the proposal of the recursive construction of the neural network in terms of modules (for unification, conjunction and disjunction operations), similar to dynamic neural module networks, and inspired by the backward chaining algorithm used for example in Prolog. Significance ------------ The creation and modeling of large knowledge bases/graphs is becoming increasingly important due to their potential support of practical tasks such as information retrieval, question answering etc. Recent efforts in distilling humanities knowledge into large-scale triple-based knowledge graphs (such as Freebase, YAGO, Google KG) are still challenged by difficulties in achieving high amounts of completeness, which is a natural product of the tremendous volume of true facts. Since hand-engineering these systems is clearly not feasible, methods capable of automatically augmenting knowledge bases/graphs based on logical reasoning and statistical inference are highly warranted. The method proposed in this paper has the potential to provide valuable insights - both for scientists as well as for practitioners - into the non-trivial task of knowledge base completion. Especially the ability of NTPs to induce human-readable, logical rules, setting it apart from the majority of previous work, may open up tremendous possibilities, particularly in the light of the recently emerging, significant interest in interpretability in AI / machine learning systems (e.g. as desired by end users, policy makers / governmental bodies and AI critics). Minor issues ------------ - the explanation of the four rules of the AND module (l. 157-162) is referring to 'line(s)' multiple times; however, the rules aren't formatted in distinct lines, so the formulation is a bit confusing; rather use 'rule(s)' instead of 'line(s)' - the last sentence of the first paragraph of section 3.4 (i.e., l. 176-177) is gramatically flawed - in l. 199, it should read 'predicates with indices r, s and t, respectively'

Reviewer 3



The paper introduces a new method for neural-symbolic reasoning on relational data by constructing neural networks that emulate the backward chaining algorithm of Prolog. In general, the paper addresses a very interesting problem, since the combination of neural networks/embedding methods and rule-based systems for reasoning on symbolic data is a promising research direction that has a long history in AI. What I found especially interesting about this method it that it allows to compute the gradient with regard to the proof success. This a very interesting property that enables, for instance, to search for rules in an ILP setting via gradient descent. It also allows to integrate this method with other gradient-based learning methods, e.g., for visual reasoning tasks. The paper is written well and reasonably good to understand. However, the description of the modules and the construction of a full NTP could be described in more detail (e.g. via additional pseudocode in the supplementary material). Currently, it would be challenging to exactly reproduce the model only from the information in the paper. Overall, there are many aspects that I like about the paper and the ideas that it pursues. My relatively low-score stems from the following concerns: First, I agree that the method could be used for the task of theorem proving. Unfortunately, in its current form, the paper actually doesn't show this. The experimental evaluation is done for simple link prediction tasks and, in addition, the paper shows qualitative results with regard to rule induction. However, in both cases no real theorem proving is required. Based on these experiments, I don't see how it can be claimed that the proposed method is a functioning theorem proving system. I'd like to emphasize that this concern is not related to the method itself, but to the generality in which it is currently presented. My second concern is the intended use of NTPs. If we are only concerned about link prediction (i.e. what is evaluated in the experiemnts), using NTPs would introduce a lot of heavy machinery but the improvements compared to a much simpler method such as ComplEx are seemingly small. Moreover, due to the large computational cost, NTP are limited to small datasets while methods like ComplEx scale very well. Furthermore, with regard to rule induction it would be important to compare the rules that are learned with NTP to those that can be extracted from embeddings or graph based methods (e.g. [1,2,3]) or classical methods ILP such as FOIL. NTPs seem to be a powerful method, but unfortunately its full benefits are currently not really tested or evaluated. Overall the paper introduces interesting ideas and a promising approach for neural-symbolic reasoning. Due to this I am currently leaning towards accept. However, I'd ask the authors to address my above concerns in their response. [1] Yang et al. "Embedding entities and relations for learning and inference in knowledge bases", ICLR. [2] Lao et al. "Random Walk Inference and Learning in A Large Scale Knowledge Base", EMNLP, 2011. [3] Neelakantan et al. "Compositional Vector Space Models for Knowledge Base Completion", EMNLP, 2015.